# A phylogenetically-conserved axis of thalamocortical connectivity in the human brain

Stuart Oldham [1,2] ✉ & Gareth Ball[1,3]

The thalamus enables key sensory, motor, emotive, and cognitive processes via connections to the cortex. These projection patterns are traditionally considered to originate from discrete thalamic nuclei, however recent work showing gradients of molecular and connectivity features in the thalamus suggests the organisation of thalamocortical connections occurs along a continuous dimension. By performing a joint decomposition of densely sampled gene expression and non-invasive diffusion tractography in the adult human thalamus, we define a principal axis of genetic and connectomic variation along a medial-lateral thalamic gradient. Projections along this axis correspond to an anterior-posterior cortical pattern and are aligned with electrophysiological properties of the cortex. The medial-lateral axis demonstrates phylogenetic conservation, reflects transitions in neuronal subtypes, and shows associations with neurodevelopment and common brain disorders. This study provides evidence for a supra-nuclear axis of thalamocortical organisation characterised by a graded transition in molecular properties and anatomical connectivity.

The thalamus is central to the structure and function of the mammalian brain. Enabled by widespread connections to the cortex, basal ganglia and the peripheral nervous system, the thalamus is engaged in a range of functions from sensory processing and motor control, through to attention and memory[1–5]. Traditionally, this breadth of functionality has been assigned to the diverse nuclear structure of the thalamus, with around 50–60 distinct nuclei gathered into 4–6 functional groups and defined on the basis of cytoarchitecture and patterns of anatomical connections[6–8].

While efforts to understand the organisation and function of the thalamus have often focused on the properties of its nuclei, a consensus on a precise nomenclature and nuclear grouping remain elusive[9]. Further, while the specificity of projections from certain thalamic nuclei to distinct cortical targets is clearly evident, not all thalamic connections are circumscribed by nuclear boundaries[10,11]. Neurons within thalamic nuclei may project to multiple cortical regions and their patterning of cortical projections can overlap significantly[9,10,12]. In contrast to nuclear divisions, recent evidence from single-cell RNA sequencing (RNA-seq) studies has refocused attention on the significant cellular diversity present within the thalamus as a potential substrate for functional diversity[13,14]. While distinct subclasses of neurons in the thalamus have long been recognised[10,15,16], the degree of cellular heterogeneity recently revealed through genetic profiling of both cortical and subcortical structures is a striking and potentially defining feature of mammalian brain organisation[17,18].

At the cellular level, neuronal subtypes can form subunits supporting a diversity of functions, the spatial arrangement of which are dictated by organisational molecular gradients[14,19–21]. In the thalamus, as with other cortical and subcortical structures, early circuit formation is scaffolded by the differential areal patterning of morphogenetic gradients during development[22–24]. These early developmental sequences are reflected by concerted variation of structural and

[1]Developmental Imaging, Murdoch Children's Research Institute, Parkville, VIC, Australia. [2]The Turner Institute for Brain and Mental Health, School of Psychological Sciences and Monash Biomedical Imaging, Monash University, Clayton, VIC, Australia. [3]Department of Paediatrics, University of Melbourne, Parkville, VIC, Australia. ✉e-mail: stuart.oldham@mcri.edu.au

functional properties along spatial axes in the adult cortex and subcortex[25–32]. Indeed, studies have found evidence for gene expression gradients and variations in cytoarchitecture both across and within thalamic nuclei[13,14,20,21,33].

In the mouse, Phillips et al. observed that thalamic nuclei are arranged along an axis of gene expression that runs in a medial to lateral direction[14]. The position of neuronal subtypes along this axis was associated with continuous variations in gene expression accompanied by differences in cortical projections, axonal morphology, and laminar targets, as well as electrophysiological properties. This work highlights how delineation of spatial gradients can give key insight into underlying principles of thalamic organisation and function[13,14,20,22]. Preliminary evidence suggests the same medial-lateral axis of thalamic organisation extends to the human thalamus[14], however it has not been extensively characterised.

Non-invasive neuroimaging is a key tool for studying thalamocortical organisation in humans[9,34–37]. While MRI can be used to target specific subcortical nuclei with a high degree of accuracy[38,39], attempts to resolve the nuclear structure of the thalamus based on patterns of cortical connectivity or correlated BOLD activation often fail to align to previously defined cytoarchitectural boundaries[40–42]. In contrast to delineating discrete brain regions, recent efforts in neuroimaging have focused on the definition of continuous axes of spatial variation based on measures of brain microstructure, anatomy, and/or function[25–27]. This approach has proven insightful, identifying continuous and overlapping patterns of anatomical variation that converge with patterns of gene expression and ontogenetic timing in the developing brain[43–45]; align with hierarchies of cortical function[29,46], and are disrupted in neurodevelopmental disorders and psychopathology[29,30,47,48]. As such, smooth transitions in anatomy, cytoarchitecture and function supported by spatially-varying gradients of gene expression may be considered a hallmark organisational motif of the mammalian brain[25,28,32,49]. Indeed, preliminary evidence suggests that thalamic functional and structural connectivity patterns are organised along a medial-lateral gradient[35,37].

In this study, we present evidence for a supra-nuclear axis of thalamocortical organisation characterised by a graded transition in molecular properties and anatomical connectivity in humans. By performing a joint decomposition of densely sampled gene expression and non-invasive diffusion tractography in the adult thalamus, we define a principal axis of genetic variation along the medial-lateral thalamic axis that corresponds to anterior-posterior patterns of thalamocortical connectivity and electrophysiological properties of the cortex. Using a large, single-cell RNA-seq survey of the brain, we demonstrate how continuous transitions in neuronal subtypes along the medial-lateral axis reflect the developmental origins of excitatory projection and inhibitory interneurons in the thalamus. We also report associations between axis-enriched thalamic genes, and genes associated with neurodevelopment and common brain disorders, and test the phylogenetic conservation of the principal axis through comparison to patterns of neuronal tracing in the adult mouse. Taken together, this study highlights an organisational axis in the thalamus, that exists across classical nuclear boundaries, is conserved across species and associated with distinct anatomical, electrophysiological, molecular, and developmental properties.

## Results

### A principal axis of thalamocortical connectivity is present in humans and conserved across species

We hypothesised that a primary organisational axis of thalamocortical connectivity exists in the human brain that spans specific nuclear boundaries and is demarcated by patterns of thalamic gene expression. We performed an unsupervised joint decomposition of postmortem gene expression[50] ($n = 3702$ samples across six donor brains[51]; 2228 genes enriched in brain tissue[52,53]) and non-invasive estimates of

cortical connectivity to 250 cortical regions (averaged over $n = 74$ healthy adults aged 22 to 36 years) from 921 thalamic seed points which we had determined to be consistently aligned across individual and/or had transcriptomic information available (see Methods; Fig. 1; Supplementary Data 1). This procedure resulted in a set of overlapping yet orthogonal components in the thalamus that sum together to reconstruct the full data matrix. Each component is represented by a set of PC scores, one per thalamic seed, defining the dominant axes of variation in both gene expression and cortical connectivity across the thalamus and a set of PC loadings that capture how strongly connections to particular cortical regions and expression of particular genes contribute to the component (Supplementary Data 1).

The principal component (PC1) represents the primary source of variation in gene expression and thalamocortical connectivity across thalamic seeds, accounting for 30.3% of variance in total (Supplementary Data 2). The spatial projection of PC1 scores varied primarily along the medial-lateral axis of the brain (Fig. 2a) with the PC1 score of each thalamic seed encoding position along the $x$-axis of MNI standard space (Pearson's $r(919) = 0.83, p = 2.83 \times 10^{-239}$, confidence interval (CI) $= [0.81, 0.85]$, two-tailed) more closely than the other Cartesian axes (anterior-posterior: Pearson's $r(919) = 0.46, p = 2.41 \times 10^{-45}$, CI $= [0.41, 0.51]$, two-tailed; dorsal-ventral: Pearson's $r(919) = -0.39, p = 2.37 \times 10^{-34}$, CI $= [-0.44, -0.33]$, two-tailed; Fig. S1). The second and third principal components (PC2 and PC3) explained less variance overall (22.4% and 13.5%) and were aligned along dorsal-ventral and anterior-posterior axes, respectively (Fig. S1; Fig. S2a, b). We conducted a series of sensitivity analyses and found that this characterisation of the principal axis was largely unaffected when applying alternative, nonlinear decomposition techniques[26,54], using a different subset of genes[14], or performing the decomposition using only gene expression or connectivity data (Figs. S3–S4). As only 921 of the 1348 seeds with transcriptomic information had been used (as the remaining 427 were judged to be inconsistently aligned across individuals), this limited our spatial coverage of the thalamus, particularly in anterior-medial and extreme lateral areas (Fig. S5). To check if this influenced our results, we also performed the decomposition on the full set of possible seeds which increased coverage of lateral areas. Results using these 1348 seeds were quantitatively similar to the main findings (Fig. S6). To further check our result, we performed bootstrapping across individuals, as well as leave-one-out cross validation across seeds, which showed our result was highly stable (Figs. S7–S8).

Graded variation in patterns of connectivity and gene expression along the medial-lateral thalamic axis is consistent with previous findings in the mouse[14]. To replicate this finding more directly, we repeated our analysis using the same framework applied to anatomical tract tracing data and transcriptomic data for 447 genes from the Allen Mouse Brain Atlas (AMBA; Fig. 2b; Supplementary Data 3)[55–58]. This revealed a primary component for the mouse data (mPC1) that explained 30.6% of variance in total (mPC2: 13.50% explained, mPC3: 9.20% explained; Supplementary Data 4) and was preferentially aligned along a medial-lateral orientation (Fig. S9a–c), as in the human. mPC1 scores correlated with the $x$-coordinate in standard CCFv3 space (Pearson's $r(33) = 0.58; p = 0.0003$, CI $= [0.30, 0.76]$, two tailed), though with some divergence in nuclei of the ventral thalamus (reticular nucleus, ventral division of the lateral geniculate)[7] and in the medial geniculate body (Fig. S10). mPC2 also varied along a medial-lateral thalamic axis but with a sizeable anterior-posterior orientation as well, while mPC3 was primarily oriented dorsal-ventrally (Fig. S9d–i; Fig. S11a, b).

PC1 loadings and mPC1 loadings of homologous genes common to both datasets ($n = 212$; Supplementary Data 5) were also highly correlated (Pearson's $r(210) = 0.63, p = 1.29 \times 10^{-24}$, CI $= [0.54, 0.70]$, two tailed; Fig. 2c). We confirmed these observations in an independent mouse dataset[14], finding highly correlated PC1 loadings of homologous mouse and human genes in the thalamus (Fig. S12). Gene

loadings in mPC2 or mPC3 showed considerably weaker relationships with the first three human PC gene loadings, indicating that mPC1 appeared to be most strongly conserved across species (Fig. S13).

Through detailed mapping of feedforward and feedback thalamocortical connections in the mouse, a recent study defined a detailed model of organisational hierarchy across thalamic nuclei[59]. Using these data, we find that medial-lateral position, defined by mPC1 scores, was significantly correlated with position in the hierarchy of feedforward–feedback interareal laminar projections[59] (Fig. 2d), providing evidence that medial-lateral orientation reflects key organisational properties of the thalamus. This relationship with organisational hierarchy appeared to be unique as neither mPC2 nor mPC3 showed a significant correlation (Fig. S14).

## Cortical patterning of thalamic connectivity follows key patterns of functional organisation

The thalamus has topographical projections to the cerebral cortex[7]; therefore, we reasoned that connections seeded along the medial-lateral gradient of the thalamus would vary along a corresponding spatial gradient in the cortex. To test this, we plotted the human PC1 loadings for each cortical region onto the cortical surface, revealing an anterior-posterior gradient (Fig. 3a). We also found these results were consistent when using a functionally driven parcellation[60] (Fig. S15). Anterior cortical regions were negatively loaded, displaying preferential connectivity to medial thalamic regions, posterior regions

were positively loaded with preferential connectivity to lateral regions (Fig. 3b). Using the mPC1 loadings from the AMBA mouse data, we observed a similar projection pattern which varied from somatomotor regions, to visual, and then to frontal/lateral cortex (Fig. 3c), following a hierarchical gradient[28,59]. Cortical loadings for mPC2 varied from the auditory, to visual/medial, and then to somatomotor and prefrontal areas, while mPC3 varied from the auditory-lateral cortex to medial/visual areas (Fig. S11c, d).

Recently, large-scale spatial gradients have been used to frame variation of a range of microstructural, connectomic, and functional properties across the cortex and subcortex[25–27]. Given the prominence of features which follow an anterior-posterior spatial arrangement in the cortex[29,46,61], we expected that PC1 loadings would vary in parallel with other cortical properties. To test this, we compared PC1 loadings to 72 cortical feature maps from the neuromaps toolbox[61]. Statistical significance was established using spin-tests (see Methods) to ensure observed correlations were not induced by low-order spatial autocorrelations[62]. Significant associations between cortical PC1 loadings were observed with several markers of functional organisation, including the primary functional gradient[26], sensorimotor-association axis[30], electrophysiological properties, and several neurotransmitter gradients (Fig. S16). The strongest association was observed with neuronal intrinsic timescales defined using MEG (Pearson's $r(248) = -0.79$, $p_{spin} = 0.0002$, CI = [−0.84, −0.74], two-tailed; Fig. 3d), with projections from the medial thalamus preferentially connected to

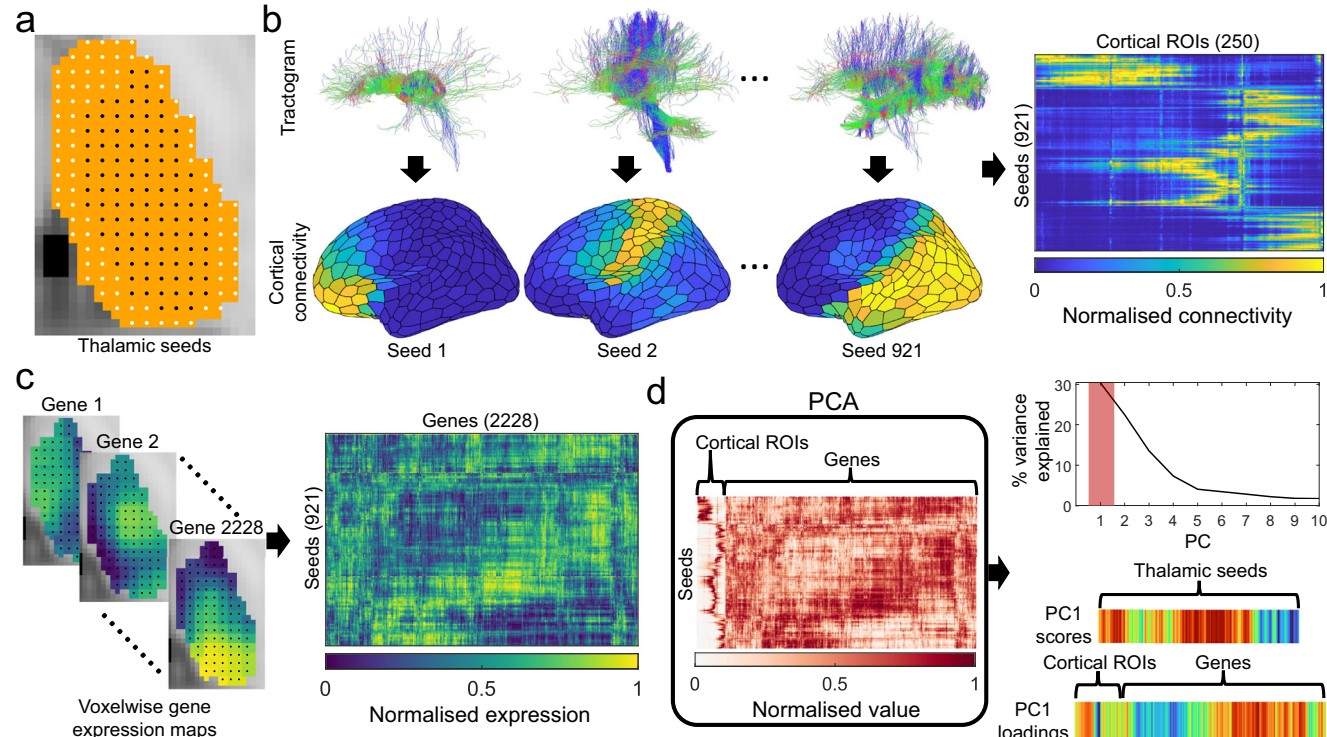

**Fig. 1 | Schematic of methodology. a** Defining thalamic seeds. Throughout the thalamic volume (orange area), a set of seeds 1.75 mm apart are defined. Only those which were consistently localised (see *Methods* for details) across participants were used as seed points (black: consistent seeds; white: inconsistent seeds). **b** Thalamic seed connectivity. Probabilistic tractography was conducted from each seed to 250 left hemisphere cortical targets based on a random parcellation. Connectivity between thalamic seeds and cortical regions was averaged across participants to produce a 921-by-250 seed-by-cortical target matrix of thalamocortical connectivity. Connectivity to cortical regions was scaled to the unit interval using a sigmoid transformation. **c** Assigning transcriptomic data to thalamic seeds. Voxelwise estimates of post-mortem gene expression for 2228 genes with differential expression in brain tissue were extracted for the thalamus. For each gene, each seed

point is assigned the expression value of the voxel it is located within to produce a 921-by-2228 seed-by-gene matrix. As above, each gene's expression levels were normalised to the unit interval according to a scaled sigmoid. **d** Joint decomposition. The seed-by-cortical connectivity and seed-by-gene matrices were concatenated and decomposed into a set of orthogonal factors by Principal Component Analysis (PCA). From the resulting principal components (PCs), the first PC (PC1) explained 30.3% of the variance in the concatenated data matrix. For each PC, the scores, one per thalamic seed, describe the representation of each component in the thalamus and the loadings, describe the contribution to the PC of connectivity strength and gene expression level for each of the cortical regions and genes, respectively.

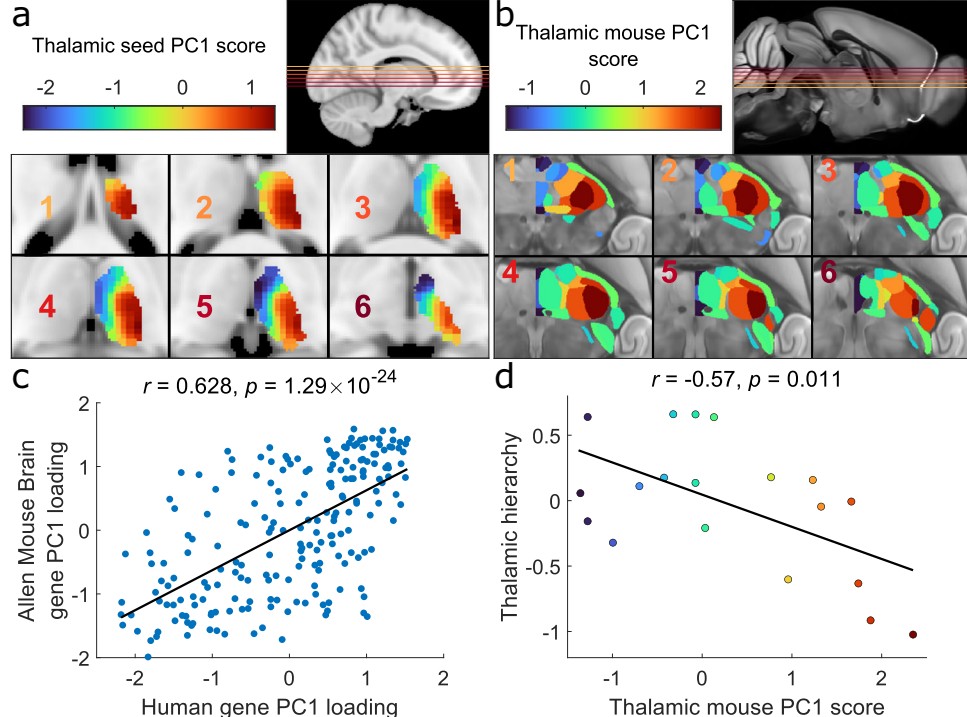

**Fig. 2 | The principal thalamic axis runs medial-laterally in the human and mouse. a** Projection of PC1 scores of the human data onto thalamic voxels. PC1 scores for each seed are projected onto the closest voxels in the thalamic mask, overlaid on six axial sections (inset; the colour of the number corresponds to the slice in the insert). PC1 scores vary along a medial-lateral direction. **b** Projection of PC1 scores of the Allen Mouse Brain Atlas (AMBA) data (mPC1) onto respective thalamic nuclei shown on six axial sections (inset; the color of the number corresponds to the slice in the insert). Note that while axonal-tracing data was obtained in the right hemisphere, we project the PC1 scores onto the left to enable straightforward comparison with the human data. **c** Scatter plot showing the relationship between PC1 loadings for homologous genes in the human and AMBA datasets (Pearson's $r(210) = 0.63$, $p = 1.29 \times 10^{-24}$, CI = [0.54,0.70], two tailed). **d** Relationship between mouse thalamic nuclei PC1 (mPC1) score and a measure of hierarchical organisation[59] (Pearson's $r(17) = -0.57$, $p = 0.011$, CI = [−0.81, − 0.17], two-tailed). Source data are provided as a Source Data file.

regions with slower cortical intrinsic timescales compared to the lateral thalamic regions.

The cortical projections of PC2 varied along a sensorimotor-frontal axis (Fig. S2c). For some measures of the cortical hierarchy, such as T1:T2 ratio, sensorimotor-association axis, and primary functional gradient, PC2's cortical projections were more highly correlated than PC1 (Fig. S17). However, PC1 was more strongly correlated with electrophysiological measures, including the intrinsic timescale. Furthermore, the cortical projections of PC3 showed an anterior/dorsal-posterior/ventral orientation (Fig. S2d). In general, PC3 was highly correlated with some electrophysiological properties (low gamma and theta power), but otherwise showed weaker relationships with certain functional gradients and neurotransmitter distributions (Fig. S18).

We performed a similar analysis in the mouse, observing that cortical mPC1 loadings were correlated with multiple properties that together characterise cortical hierarchical organisation (Fig. 3e; Fig. S19). This relationship with measures of the cortical hierarchy appears to be unique to mPC1 as both mPC2 and mPC3 showed weaker relationships across all the cortical properties tested (Figs. S20–S21).

## Cellular and molecular composition varies as a function of the medial-lateral thalamic axis

Neuronal cell types are distributed non-uniformly in the thalamus[63,64] with distinct cellular subtypes differentiated by graded variations in gene expression both across and within discrete thalamic nuclei[13,14,20]. Therefore, we hypothesised that cellular composition, evidenced by differential gene expression, would vary systematically along the medial-lateral axis. Using genes with the largest positive and negative PC1 loadings in the human data (Supplementary Data 6; $n = 100$ each;

after accounting for spatial autocorrelation across thalamic seeds using spin tests; see Methods[65]), we queried a comprehensive Drop-seq analysis of 89,027 cells from the adult mouse thalamus[33] to test if genes with medial-lateral patterns of expression were enriched for different thalamic cell type markers.

Both lateral- ($n = 100$) and medial-genes ($n = 100$) were significantly enriched for neuron class markers (lateral-genes: enrichment = 6.57, $p_{FDR} = 3.44 \times 10^{-28}$; medial-genes: enrichment = 2.09, $p_{FDR} = 0.02$); while lateral-genes were additionally enriched for oligodendrocyte markers (enrichment = 8.03, $p_{FDR} = 2.28 \times 10^{-11}$), and medial-genes for glial (astrocyte: enrichment = 3.39, $p_{FDR} = 0.007$; ependymal: enrichment = 5.96, $p_{FDR} = 1.12 \times 10^{-11}$) markers (Fig. 4a; Supplementary Data 7). To further differentiate between medial and lateral gene sets, we focused on neuronal classes and tested enrichment of three previously identified neuronal subtypes[33]: Rora (excitatory neurons expressing Slc17a6), Gad2/Ahi1 (neurons largely expressing inhibitory markers Gad1 and Gad2), and Habenula (cholinergic and glutamatergic neurons in the habenula). Lateral- and medial-genes showed differential patterns of enrichment, with genes expressed by Rora subtypes and enriched in lateral genes (enrichment = 8.64, $p_{FDR} = 3.11 \times 10^{-26}$) encoding glutamate receptors (Grid1/Grm1), voltage-gated channels (Scn1b/Kcna2/Scn8a) and calcium transporters (Slc24a2) with medial-genes where enriched for Habenula (enrichment = 5.81, $p_{FDR} = 7.41 \times 10^{-6}$) and Gad2/Ahi1 (enrichment = 8.19, $p_{FDR} = 1.12 \times 10^{-7}$) makers, including those for forebrain interneurons (Dlx1[66]) and GABAergic neurons (Cnr1; Fig. 4b; Supplementary Data 7).

Each neuronal class comprised several closely related clusters revealed through two-dimensional embedding of gene expression in each subtype[33]. Though distinct in identity, the close proximity of

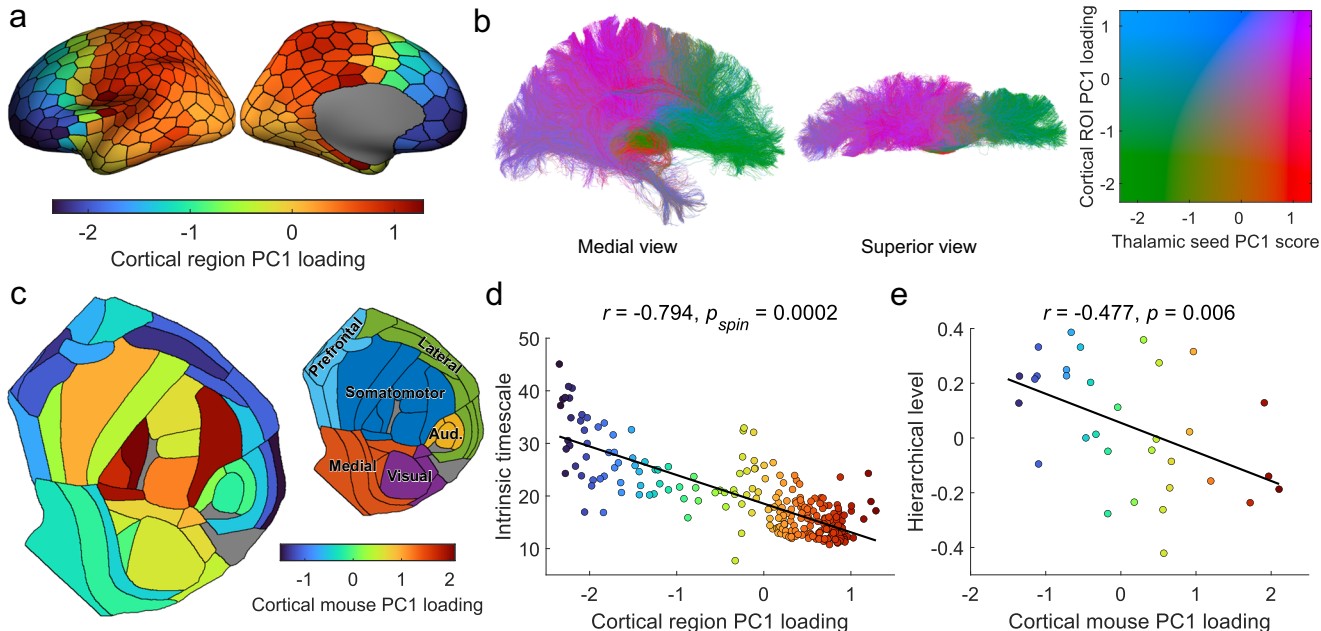

**Fig. 3 | Cortical patterning of thalamic connections corresponds to key cortical gradients. a** The PC1 loadings for cortical regions are shown projected onto the cortical surface, revealing an anterior-posterior gradient of thalamocortical connectivity. **b** Representative tractogram (plotted in MNI152 space) where each streamline is coloured according to its PC1 score and loading of the corresponding seed point and cortical region it traverses between. **c** Projection of PC1 loadings of the Allen Mouse Brain Atlas (AMBA) data onto respective cortical regions, displayed as a flat map. The smaller flat map plot indicates major cortical divisions (prefrontal, lateral, somatomotor, visual, medial, and auditory)[59]. Grey regions indicate cortical areas which no gene expression and/or connectivity data was available. **d** Correlation between cortical region PC1 loadings (for the human data) and MEG intrinsic timescales (Pearson's $r(248) = -0.79$, $p_{spin} = 0.0002$, CI = [−0.84, −0.74], two-tailed). Points are coloured according to their PC1 loading. **e** Correlation between mouse cortical PC1 loadings and hierarchical level (Pearson's $r(30) = -0.48$, $p = 0.006$, CI = [−0.71, −0.15], two-tailed). Points are coloured according to their mouse cortical PC1 loading. Source data are provided as a Source Data file.

neuronal classes/subtypes, and clusters within these classes (which we refer to as subclusters), suggested overall similar patterns of gene expression across cells with graded transitions across borders between both clusters and neuronal class[13,33]. Based on this, we reasoned that a graded pattern of gene enrichment would be evident across adjacent subclusters within each neuronal class, rather than discrete clusters with or without enrichment. We repeated the overrepresentation analysis focusing on neuronal cell subclusters (Supplementary Data 7), finding the degree of medial and lateral enrichment transitioned smoothly across subclusters (Fig. 4c), thereby encoding position of individual cell types across the thalamus. This smooth transition across subclusters is evident when projecting enrichment ratios onto embedded cells (Fig. 4d), and supports previous work demonstrating that cell types near the border of adjacent thalamic nuclei can display similar patterns gene expression to each other and do not belong to separate populations, although for certain sub-regions clearer boundaries can be observed[14].

As with PC1, we identified the top 100 genes associated with the most negative and positive PC2 and PC3 loadings (Supplementary Data 6) were also enriched for several cell-classes. For PC2, ventral-genes were enriched for glial markers (astrocyte: enrichment = 4.62, $p_{FDR} = 3.34 \times 10^{-5}$; polydendrocyte: enrichment = 4.56, $p_{FDR} = 0.004$; oligodendrocyte: enrichment = 7.75, $p_{FDR} = 2.27 \times 10^{-10}$), while dorsal-genes were enriched for neurons (enrichment = 7.78, $p_{FDR} = 1.83 \times 10^{-32}$; Fig. S22a; Supplementary Data 8). In PC3, both anterior- and posterior-genes were enriched for glial (anterior astrocyte genes: enrichment = 3.60, $p_{FDR} = 0.02$; posterior astrocyte genes: enrichment = 3.82, $p_{FDR} = 0.02$; anterior oligodendrocyte genes: enrichment = 5.02, $p_{FDR} = 0.004$) and neuronal markers (anterior-genes: enrichment = 2.45, $p_{FDR} = 0.03$; posterior-genes: enrichment = 3.56, $p_{FDR} = 1.89 \times 10^{-4}$; Fig. S23a; Supplementary Data 9). PC2, but not PC3, also showed a graded transition across neuronal subclusters, although this

transition was less apparent across Gad2/Ahi1 than was seen with PC1 (Figs. S22c, d, S23c, d).

## Genes expressed along the medial-lateral axis are associated with thalamic development and disease

Forebrain development is founded on early canonical molecular gradients during gestation, which are reflected in the differential timing of key developmental processes across structures[24,32]. Spatial gradients in adult gene expression data vary along developmental axes[29] and recent evidence has shown transcriptional profiles of neurons retain a persistent marker of their developmental origins[67]. Based on this evidence, we examined if the medial-lateral axis captured differences in developmental timing across the thalamus. Using a database of post-mortem RNA-seq data acquired from multiple brain regions across the human lifespan[68], we identified a set of genes differentially expressed over nine developmental windows (Methods). We observed divergence across the medial-lateral axis, with medial genes enriched during both the prenatal and postnatal developmental periods, while lateral genes are largely enriched postnatally (Fig. S24). Differential expression of several prenatally-enriched genes was identified across the medial-lateral axis (Supplementary Data 10), with both medially- and laterally-enriched genes involved in thalamocortical outgrowth (DSCAML1[69], SLIT1[70], and FZD3[71,72]), while medially-enriched genes were also involved in interneuron migration from the forebrain ganglionic eminence (DLX1[66] and GLRA2[73]) and lateral enrichment was linked to functional maturation of synapses (ADCY1[74]). PC2 also showed differential expression across development (Fig. S25; Supplementary Data 10), as did PC3 to a lesser extent (Fig. S26; Supplementary Data 10).

Multiple neurodevelopmental and neurodegenerative disorders are associated with thalamic dysfunction[75,76], therefore we examined if genes expressed along the medial-lateral axis were also differentially

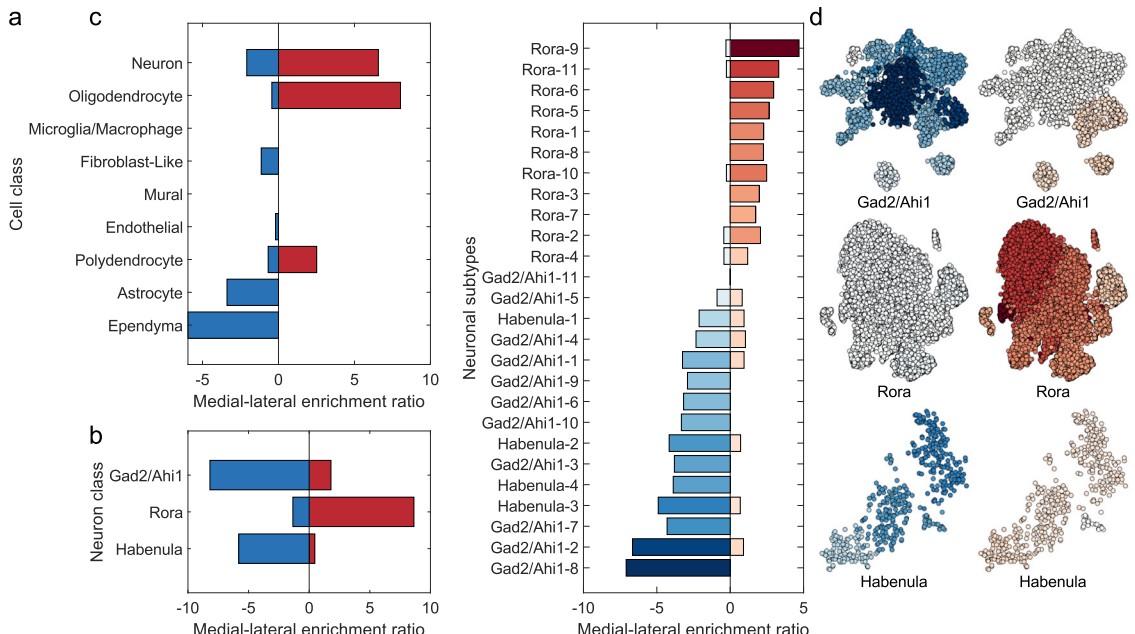

**Fig. 4 | PC1 reflects graded distinctions in cytoarchitecture. a** Enrichment ratio for medial- and lateral-genes expressed by cell class. The medial-lateral enrichment ratio is positive when that cell class is enriched for lateral-genes and negative when enriched for medial-genes. **b** Enrichment ratio for medial- and lateral-genes expressed by each neuron class[33]. **c** Enrichment ratio for medial- and lateral-genes expressed by each neuronal subcluster. Subclusters were defined by clustering cells within each neuron class by their pattern of gene expression. In the bar plot, neuronal subclusters are ordered by their summed medial-lateral enrichment ratio. **d** Enrichment ratios projected onto t-SNE plots for the different neuron type sub-clusters. Cells belonging to each subcluster are coloured according to their enrichment value as indicated in (**c**) for medial- (left) and lateral-genes (right). Source data are provided as a Source Data file.

associated with disease. We found that medial-genes were significantly enriched ($p_{FDR} < .05$) for genes associated with alcohol-related disorders (alcoholism, alcoholic intoxication, alcohol withdrawal seizures), eating-related disorders (eating disorders, hyperphagia, and anorexia), and psychosis-related disorders (psychotic disorders, schizoaffective disorder, and schizophrenia; Supplementary Data 11). Genes encoding for gamma-aminobutyric acid (GABRB1/GABRG1/GABRG3), cannabinoid (CNR1), serotonin (HTR2R/HTR2C), and neuropeptide (NPY2R/NPY5R) receptors, in addition to oxytocin (OXT), were commonly associated with the disorders showing enrichment for medial-genes. For lateral-genes, these showed enrichment for epilepsy-related (generalised and myoclonic), bipolar disorder, schizophrenia and other generalised movement/neurological conditions (Supplementary Data 11), with genes for sodium/potassium transport and channels (ATP1A3/KCNA1/KCNA2/KCNC1/SCN1B/SCN8A), and glutamate receptors (GRM1/GRM4) implicated across disorders.

PC2 and PC3 also showed significant enrichment for several disorders. PC2 was enriched for mood, substance abuse, and psychosis disorders along its negative thalamic axis (dorsal-to-ventral), with genes for gamma-aminobutyric acid (GABRA4/GABRA5), corticotropin releasing hormone (CRH), opioid receptors (OPRM1), and glutamate metabotropic receptors (GRM7) being implicated across disorders (Supplementary Data 12; the positive axis of PC2 did not show any significant enrichment). The positive axis of PC3 (varying anterior/medial-to-posterior) also was primarily enriched for mood disorders, and was linked to serotonin related genes (HTR1A/HTR2B/TPH2; Supplementary Data 13; the negative axis of PC3 did not show any significant enrichment).

## Discussion

The thalamus and its reciprocal cortical connections are crucial to shaping whole brain neural activity, yet the underlying organisational principles of these connections are not well understood[7,13,14]. In this study, we have defined a principal organisational axis of the adult human thalamus, characterised by a graded transition in aggregated gene expression, that is conserved across species and aligned to functional cortical organisation via a medial-lateral to anterior-posterior mapping.

Previous work identified an axis of gene expression running in a medial-lateral orientation at the level of individual thalamic nuclei in the mouse[14]. We extend these observations, demonstrating continuous variation in structural and molecular properties along this spatial axis. By decomposing bulk tissue microarray data into constituent cell types using markers from a comprehensive single-cell database[33], we identify potential differences in cell class distribution across the principal thalamic axis. Lateral areas, adjacent to the white matter, were enriched for oligodendrocyte markers, while medial areas, adjacent to ventricles and developmental sources of radiating glial cells, were enriched for other glial markers. Similar divergence was observed across neuronal classes suggesting that composition of neuronal populations also varies along the thalamic axis, with lateral areas preferentially enriched for excitatory markers, and medial areas for inhibitory ones.

Traditionally, the identity and functional role of thalamic nuclei have been in part determined through examination of afferent connections and cortical projection targets[1,7]. We examined patterns of cortical connectivity with the thalamus using non-invasive diffusion tractography, validating our findings using anatomical tracing data in the mouse. Cortical projections from the medial-lateral thalamic axis mapped to an anterior-posterior cortical gradient, aligning with cytoarchitectural and function markers of cortical hierarchy[25,29,32,46]. We find this cortical projection gradient mirrored variation in neuronal intrinsic timescales, fundamental patterns of oscillatory activity and neural dynamics in the cortex[77]. This cortical patterning is likely conserved across species. The gradient of thalamic projections in the mouse cortex was also related to measures of cortical hierarchies[28,59] and the thalamic medial-lateral to cortical anterior-posterior mapping has been observed in tract-tracing experiments in rodents and primates[78,79]. In the mouse, electrophysiological properties of neurons including action potential threshold/width, ion channel and receptor

profiles vary as a function of position along the primary axis[14]. We additionally find that medial-lateral position encodes flow of hierarchical information across the mouse connectome[59], further highlighting potential functional importance of the thalamic axis. This relationship between measures of cortical hierarchy and thalamic cortical projections appeared distinct to PC1 (i.e., the medial-lateral axis) in the mouse as compared to other thalamic axis, but in the human data PC2 (corresponding to a dorsal-ventral thalamic axis) also showed a clear relationship with these measures, which may reflect phylogenetic differences. Mammalian species can differ in their thalamic composition, for example having different proportions and distributions of inhibitory neurons amongst thalamic nuclei[63,80] and this likely translates to functional differences, however the patterning of these distributions may remain phylogenetically conserved[63]. Overall our results indicate a close alignment of thalamocortical connectivity with cortical organisational principles and suggest a conserved thalamus-to cortical mapping across species.

Prior studies have observed that measures of functional connectivity between the thalamus and cortex in humans form large-scale gradients[37] that partially align with spatial distribution of core/matrix neuronal cell types and are correlated with timescales of neural activity in the cortex[81]. Electrophysiological signals in the cortex are shaped by a balance of excitatory and inhibitory inputs, which the thalamus is involved in the regulation of[82]. Systematic variation in thalamic electrophysiological properties also influence functional activity in the cortex[11,14]. Taken alongside previous findings, our work supports the notion that intrinsic cortical dynamics, at least in part, are constrained by a principal mapping of thalamic projections to the cortex, potentially underwritten by distributional variations of specific neuronal populations[13] along the medial-lateral axis.

Expression gradients across and within nuclei may offer a more parsimonious account of function attributed to thalamic nuclei[13]. However, while prominent, several aspects of thalamic organisation remain unaccounted for by the medial-lateral axis. Different projection systems of the thalamus have distinct patterns of gene expression[14] and cytoarchitectonic boundaries between nuclei are well-documented[7,13]. As an example, in the mouse data, we found the reticular nucleus and geniculate bodies diverged from the primary axis. The reticular nucleus and ventral division of the lateral geniculate form part of the ventral thalamus, or prethalamus, a structure distinct from the rest of the (dorsal) thalamus in terms of development, connectivity and function[7,22,83]. In our human data, however, due to the position at the extreme lateral border of the thalamus, these structures were only sparsely sampled and were unable to be fully characterised. Similarly, anterior-medial areas were also not fully represented in our analysis, and so future research should assess if the thalamic gradients fully extend to these areas as well.

We recognise that our estimates of connectivity along thalamic gradients may be affected by biases inherent to diffusion tractography including difficulty tracing from deep thalamic regions and the potential for false positive or negative connections. We note that tractography-derived thalamocortical connectivity broadly aligns with tract-tracing findings in primates[34,84,85], and we have attempted to mitigate this risk through validation of our human tractography results to those obtained using gold-standard tract-tracing in the mouse. Thus, while some finer details may be missed with tractography, we would expect the major patterns of connectivity we have described to be robust to specific tracking limitations. Nevertheless, addressing these limitations through new tractography techniques, improved MRI acquisitions, or comparison to thalamocortical gradients in non-human primates using tract-tracing data represents an exciting direction for future research.

We note that, as with cortical arealisation, the presence of supra-areal organisational gradients does not preclude functional localisation to discrete regions, nor does it dismiss clear neurobiological differences between nuclei[25]. Localisation of functions to discrete nuclei is well supported by an extensive body of evidence. Specific thalamic nuclei are directly involved in specific cognitive[8], sensory[86], and motor[87] activity and display preferential connectivity to focal cortical targets[7]. However, these focal projections vary in a spatially systematic way which we and others have observed[14,35,37] and the notion of discrete thalamic units cannot account for all functional heterogeneity observed in the thalamus. Similarly, thalamic nuclei are clearly distinguishable by certain neurobiological properties, implying a degree of specificity, but in other features they show continuous variation. Therefore, discrete and continuous organisational principles of the thalamus are not mutually exclusive, and may indeed be complementary. It has been noted that different theories of how neurons within specific nuclei are recruited to enable specific functions, often require some form of intra-thalamic communication[10], therefore the cardinal gradients we observe in the thalamus may correspond to such a system. Indeed, intrinsic patterns of thalamic functional connectivity occur along continuous spatial dimensions[88], suggesting that internal thalamic activity may be shaped via connectivity along these spatial axes. Reconciling how continuous and discrete patterns of thalamic organisation interact to support whole brain dynamics, and what specific functional roles each may have, is a point of keen interest going forward.

Our results also suggest that multiple organisational axes may exist in the thalamus. When considering PC2 and PC3, we find that, much like PC1, these correspond to spatial variations along approximately cartesian planes (dorsal-ventral and anterior-posterior, respectively). Other subcortical structures, like the hippocampus and striatum, also show organisational variation along similar planes[89–95], as does the cortex[32,96–98]. The alignment of features along these axes may be reflective of molecular gradients that arise in development which shape the formation of the brain[96,99]. Indeed, work has indicated the thalamus develops along lateral-medial[100–102] and dorsal-ventral axes[103], which matches our observations of these axes show differential gene-expression across development. Thus, the alignment of thalamic neurobiological and functional properties along similar axes may be a function of developmental molecular gradients. Distinguishing how neurobiological features in cortical and subcortical structures vary along different axes will be important for understanding how shared and distinctive developmental and genetic mechanisms shape brain organisation.

We found genes expressed along the medial-lateral axis were associated with numerous disorders including eating-related, alcohol-related, psychosis, neurodevelopmental, and epilepsy related disorders, reflecting the diversity of thalamic function in human behaviour and neurophysiology. The specific genes enriched for these disorders included those which encode for GABA, cannabinoid, serotonin, glutamate, and neuropeptide receptors as well as sodium/potassium transport. Eating, alcohol-related, psychosis, neurodevelopmental and epilepsy disorders have been associated with both thalamic and neurotransmitter abnormalities[75,76,104–108]. As the thalamus has a key role in neuromodulation[109], disruptions to thalamic and/or neurotransmitter systems may be reflected by altered neurotransmitter signalling along the medial-lateral axis. Furthermore, several genes were also differentially enriched across early developmental windows, and temporal sequences of neurogenesis occur along a medial-lateral direction in the thalamus[100,101]. Thus, disruptions along the medial-lateral axis during development could cause substantial alterations in connectivity. Our study provides a potential framework for interrogating fundamental properties of thalamocortical organisation across species. Examining how principles of thalamocortical connectivity emerge and are potentially disturbed, remains an underexamined area of research which addressing would promote new insights into the course of healthy and abnormal development.

In summary, we find a principal axis of transcriptomic and structural connectivity in the human thalamus that is situated along a medial-lateral axis and conserved across species. Position along this axis encodes functional hierarchy, variations in cellular composition, and is mapped to key properties of cortical function, revealing a simple organisational principle of thalamocortical connectivity.

## Methods

### Human neuroimaging data

Minimally-processed structural and diffusion data from 100 unrelated participants were acquired from the Human Connectome Project[110,111]. Of these 76 (46 females, age mean ± SD: 28.39 ± 3.95 years) were retained after quality control (see below). Data were acquired on a customised Siemens 3T Connectome Skyra scanner at Washington University in St Louis, Missouri, USA. Structural T1-weighted data were acquired with 0.7 mm³ voxels, TR = 2400 ms, TE = 2.14 ms, FOV of 224 × 224 mm. Diffusion data was acquired using a multi-shell protocol for the DWI (1.25 mm³ voxel size; TR = 5520 ms; TE = 89.5 ms; FOV of 210 × 180 mm; 270 directions with $b$ = 1000, 2000, 3000 s/mm², 90 per $b$ value, and 18 $b$ = 0 volumes). Image pre-processing is described in detail elsewhere[110–112]. Briefly, diffusion data were corrected for EPI susceptibility and signal outliers, eddy-current-induced distortions, slice dropouts, gradient-non-linearities and subject motion[112]. T1-weighted data were corrected for gradient and readout distortions prior to being processed with FreeSurfer.

From the pre-processed data, the fibre orientation distributions were extracted from the diffusion data using the multi-shell multi-tissue Constrained Spherical Deconvolution algorithm in MRtrix (version: 3.0.15)[113–115]. A five-tissue-type segmentation was extracted from the T1w-weighted image[116]. To parcellate the left cortex, the cortical surface was divided 250 approximately equally sized regions using the parcellation_fragmenter tool (https://github.com/miykael/parcellation_fragmenter). A volumetric representation of this parcellation was then created using FreeSurfer (version: 5.3.0)[117]. As an additional robustness check, we also used the Schaefer 400, 17 network parcellation[60] for a supplementary analysis. These processing steps described here had been previously performed for separate studies[118,119], and this processed data was accessed on the MASSIVE high-performance computing system[120].

### Human gene expression data

Gene expression levels were assigned to thalamic seeds using high-resolution maps of estimated gene expression in the thalamus[50]. Briefly, Gaussian Process Regression was used to estimate the spatial dependence of gene expression between neighbouring locations in the cortical and subcortical structures based on post mortem microarray data from the Allen Human Brain Atlas (AHBA)[51]. The AHBA contains 3702 microarray samples of 58,692 probes across six brains. Microarray pre-processing is detailed elsewhere[50,51]. Using spatial models of gene expression estimated at discrete locations in volumetric space, voxel-wise expression maps of 18,836 genes were generated[50]. In the present study, we focused on a previously identified list of 2413 genes with differential expression in the human brain[52,53], of which 2228 were present in the high-resolution dataset (Fig. 1c) which were downloaded from an online repository (http://www.meduniwien.ac.at/neuroimaging/mRNA.html). As few samples in the AHBA were obtained in the right hemisphere, we elicited to only use data from the left hemisphere.

### Thalamic seed definition and quality control

To measure variation in connectivity and gene expression across the thalamus, we defined a set of seeds. Seeds are defined within an MNI152 thalamic mask[95] (1811 seeds total; 1.75 mm apart) and are registered to each participant using transforms (FLIRT and FNIRT) provided by the HCP[111]. This number of seeds was selected to maintain a balance between dense spatial coverage of the thalamus and minimising

computational burden. Quality control was performed by defining a binary vector indicating if a seed was inside or outside each participants own thalamic mask (defined using tissue segmentation with FSL's FIRST[121]). Participants whose vector was not highly correlated with others (mean $r$<0.7) were excluded ($n$ = 24), as these participants likely have an inconsistent spatial distribution of seeds to other participants. The choice of only including seeds which could be consistently registered across participants was done so we could reliably ensure that only thalamic areas were being sampled. Seeds that were (a) present in over 85% of the remaining participants and (b) were located within areas where transcription data was available (i.e., the seed resided in a voxel for which expression data was available) were retained (Fig. 1a). Of the 1811 possible seeds, 1348 had transcriptomic information associated with them, and 921 passed the quality control procedure outlined above. As a consequence of this, areas of the anterior-medial and lateral thalamus did not have complete coverage (Fig. S5).

### Thalamic seed connectivity

To estimate connectivity between the thalamus and cortex, for the remaining 76 participants, 5000 streamlines were generated from each of the 921 spatially-consistent thalamic seeds using the second-order integration over fibre orientation distributions tractography algorithm[114,122] (1.25 mm step size; 45° maximum angle; 0.05 fibre orientation distribution cut-off) with Anatomically Constrained Tractography (using the five-tissue-type image)[116] applied using MRtrix3 (version 3.0.15)[114] (Fig. 1b). Streamlines were assigned to the nearest left hemisphere cortical targets within a 5 mm radius of their endpoint. Cortical target regions were based on a random parcellation where each parcel had approximately equal surface area (Fig. 1b). Connectivity between seeds and cortical regions was averaged across participants to produce a 921-by-250 matrix of thalamocortical connectivity. Expression values were sampled from the voxelwise gene expression maps by assigning seeds the expression values of the voxel they resided in, producing a 921-by-2228 matrix of thalamic seed gene expression. We then concatenated these two matrices to make a single 921-by-2478 data matrix defining the cortical connectivity and gene expression across thalamic seeds.

To ensure comparison between genes and cortical connection values, these data were normalised using a scaled sigmoid transformation to the interval [0,1]. This first involved applying a sigmoidal transformation to the raw data:

$$S(x) = \frac{1}{1 + \exp\left(-\frac{x - \langle x \rangle}{\sigma_x}\right)}, \tag{1}$$

where $S(x)$ is the normalised value of a gene/connection, $x$ is the raw value, $\langle x \rangle$ is the mean and $\sigma_x$ is the standard deviation of the values of that gene/connection across thalamic seeds. Following the sigmoidal transform, each gene or cortical connection was linearly scaled to the unit interval. This transformation was used to reduce the impact of outliers in the data[55,123] (Fig. 1d). The same concatenation and normalisation procedure was applied to the mouse axonal tracing and thalamic gene expression data.

### Mouse data

We used gene expression[58] and anatomical connectivity[56] data for the mouse from the AMBA. Data processing has been detailed elsewhere[55,57]. Expression data were extracted for the 213-region mouse parcellation of ref. 56. Of the 19,417 genes for which expression had been measured across all 213 regions, we extract those for which there was (a) complete expression data for all 35 thalamic nuclei, and (b) were part of the 500 most DE genes across mouse thalamic nuclei as identified by ref. 14. This resulted in a total of 447 genes being retained, producing a 35-by-447 thalamic-by-gene expression matrix.

Connectivity data was derived from the Allen Mouse Brain Connectivity Atlas, which consists of 469 anterograde viral microinjection experiments conducted on C57BL/6J male mice at age P56[56]. We extracted connections from 35 thalamic nuclei to 38 cortical targets, producing a 35-by-38 axonal thalamic-by-cortical region connectivity matrix for use in the decomposition. For visualisation and calculation of mouse thalamic nuclei coordinates, the Common Coordinate Framework version 3 atlas for the AMBA was used[58,124].

## Joint decomposition

We decomposed the concatenated 921-by-2478 ($n \times m$) data matrix, $M$, into a set of orthogonal components using Principal Component Analysis (PCA) via Singular Value Decomposition (SVD):

$$M = USV^T, \tag{2}$$

where, $US$ is a $921 \times k$ matrix represents the Principal Component (PC) scores, one per thalamic seed for each of $k$ components; and $V$ is a $2478 \times k$ matrix representing the PC loadings, or coefficients, that denote the contributions of each cortical region's (normalised) thalamic connectivity or each gene's (normalised) thalamic expression to each component ($M$ was centred prior to SVD/PCA). The decomposition is normally truncated to $k < \min(n,m)$ and the variance explained by each component, $\lambda_k$, is given by its singular values, $s_k$:

$$\lambda_k = \frac{s_k^2}{n-1} \tag{3}$$

This approach reduces the dimensionality of the data by finding components (axes which maximise the variance explained in the data) which are orthogonal to each other. We repeated this analysis in the mouse using the corresponding, concatenated 35-by-485 data matrix (38 cortical regions and 447 genes). PCA decompositions were conducted using MATLAB 2020a. Note that all decomposition results were z-score normalised across thalamic seed scores, cortical region loadings, and gene loadings, for each component.

## Sensitivity analyses

To assess the potential impact of the decomposition approach and the data used on our observations, we performed a series of sensitivity analyses:

- Performed the joint decomposition using a nonlinear alternative to PCA (diffusion embedding using the BrainSpace MATLAB toolbox; version: 0.1.10; https://brainspace.readthedocs.io/en/latest/index.html)[26,54] to test if our observations were limited by using a linear model.
- Performed the PCA using only human homologues of genes previously identified as differentially expressed along the medial-lateral axis in the mouse[14] to test if the results were consistent when a more restricted gene-set was used.
- Performed the PCA on the connectivity and gene expression data matrices separately to test if the decomposition was driven by connectivity or gene expression.
- Performed an initial PCA on the concatenated top ten components from the separate PCA of the connectivity and gene expression data to ensure each data type contributed the same number of features to the decomposition.
- Calculated the cosine affinity matrix for the connectivity and gene expression matrices separately, and performed diffusion embedding on the averaged affinity matrices so each data type contributed equally to the non-linear decomposition.

To ensure the robustness of our analysis, we conducted bootstrapping and leave-one-out cross-validation procedures. Specifically, we performed bootstrapping by randomly selecting (with replacement) 76 individual connectivity matrices, and then averaged and normalised them before performing decomposition with and without the gene data (because the gene data cannot be bootstrapped and this may bias the decomposition on the full data, we also examined how the bootstrapping performed when the decomposition was performed just on the connectivity data). We evaluated the robustness of the iterative decomposition by comparing the variance explained for each component and the correlation between the PC1 scores and loadings between the original and the iterative decomposition across each bootstrap iteration. Additionally, we used leave-one-out cross-validation by iterating across seeds for the original concatenated group averaged connectivity and gene expression. In each iteration, we removed one seed from the concatenated matrix and performed the decomposition. We then calculated the root-mean-square-error for the PC1 scores across the remaining seeds and compared it with the corresponding scores from the original decomposition. These procedures allowed us to ensure the stability and consistency of our findings.

## Maps of human cortical properties

We obtained maps of multiple cortical features from the neuromaps toolbox (version: 0.0.3; https://netneurolab.github.io/neuromaps/)[61]. This dataset consists of 72 high-resolution maps of different cortical properties including measures of tissue microstructure, gene expression, metabolism, neurotransmitter receptor distribution, electrophysiology, and cortical expansion amongst others. All maps were transformed to the fsaverage 164 k template using neuromaps and the Connectome Workbench (version: 1.5.0; https://www.humanconnectome.org/software/connectome-workbench). Vertexwise properties were averaged within each parcel to get a single value for each region of the parcellation.

## Maps of mouse cortical properties

We used maps of nine different cortical properties of the mouse brain which have previously been found to be reflective of mouse hierarchical organisation[28]. These included the ratio of T1-weighted to T2-weighted (T1w:T2w) images, mean cell density for parvalbumin-containing (PV) cells[125], cytoarchitectonic classification based on regional eulamination[126], cortical gene expression[58], intracortical axonal connectivity[56], and inferred hierarchy from feedforward–feedback laminar projection patterns between cortical and thalamic regions[59]. Further details of how these data were extracted and preprocessed are available elsewhere[28,59].

## Additional gene expression datasets

Using the DropViz (http://dropviz.org/) database, we downloaded lists of differentially expressed genes for nine thalamic cell types: neurons, ependyma, astrocyte, polydendrocyte, endothelial, mural, fibroblast-like, microglia/macrophage, and oligodendrocyte[33]. Differentially expressed genes were defined using the following parameters: minimum fold ratio of three; maximum p-value exponent of −50; minimum log expression in target of one; and a maximum mean expression in comparison of six). Where cell types constituted more than one cluster (neuron, $n = 3$; oligodendrocyte, $n = 2$), they were combined into a single 'target' cluster and compared to all other cell types. Neuron genes consisted of three clusters/classes, Rora, Gad2/Ahi1, and Habenula, each of which had several subclusters ($n = 11$, 11, and 4 respectively). For enrichment of subclusters, each subcluster was compared to a reference set which consisted of all other subclusters in the different subtypes, and the background set was defined as all genes expressed by neurons.

Gene homologues were identified using Ensembl BioMart (https://www.ensembl.org/index.html; reference genomes: human GRCh38.p13; mouse GRCm39)[127]. Gene lists were filtered to only include: genes with identified mouse-human homologues; genes with protein expression in

the human thalamus (list from The Human Protein Atlas; version: 21.1; https://www.proteinatlas.org/)[128]; and genes with expression in the top 75% based on aggregated unique molecular identifier across cell types in the mouse thalamus, ensuring that corresponding genes are expressed in both the mouse and human thalamus.

### Overrepresentation analysis

To assess enrichment of genes across different gene sets, we used the hypergeometric statistic:

$$p = 1 - \sum_{i=0}^{x} \frac{\binom{K}{i}\binom{N-K}{n-i}}{\binom{N}{n}}, \tag{4}$$

where $p$ is the probability of finding $x$ or more genes from a gene list $K$ in a set of $N$ randomly selected genes drawn from a background set $N$. Enrichment was expressed as the ratio of the top $n$ genes present in the gene list of interest, compared to the proportion in the full background. The overrepresentation analysis for different cell types, neuron classes, and neuronal subtypes, was performed in Python (version: 3.8.5).

To identify genes associated with disorders, we used the WEB-based GEne SeT AnaLysis Toolkit (https://www.webgestalt.org/)[129], using protein encoding genes from the human genome as a reference set and the DisGeNET[130], GLAD4U[131], and OMIM[132] databases as functional sets. A Benjamini-Hochberg false discovery rate of $p_{FDR}$ <.05, with a minimum and maximum overlap of 4 and 2000 genes respectively, was used to identify significant categories.

### Spatial nulls

As cortical and subcortical features exhibit spatial autocorrelations, we implemented spatial-autocorrelation-preserving permutation tests to assess statistical significance (commonly known as "spin-tests") between pairs of brain maps and to correct for smooth spatial autocorrelation in thalamic maps of gene expression. We conducted two separate spin-tests, one for the cortex and the other for the thalamus.

A cortical spin-test[62] was used to find which of the cortical maps from the neuromaps toolbox was significantly correlated with the PC1 loadings in the cortex. First the centroid of each cortical region on the FreeSurfer spherical projection was found. These coordinates are then rotated at three randomly generated angles. The Euclidean distance between each pair of rotated region centroids and original region centroids is the calculated. Regions are then iteratively assigned to rotated ones by finding which rotated region is closest on average to all original regions, and then mapping that rotated one to the most distant original region. This process is repeated until each rotated region is mapped to a unique original one. Based on this mapping, regional values can be mapped to a new region to preserve spatial contiguity. This procedure was repeated to produce 10,000 permutations. We calculated a spin-test derived $p$-value ($p_{spin}$) for a pair of brain maps by comparing the Pearson correlation between them to a distribution of correlations between one empirical map and 1000 spatial permutations of the other (this was repeated such that each brain map was permuted and compared to the empirical pair, the mean of these runs $p$-values was then taken)[62]. Significance was determined at $p_{spin}$ < 0.05.

To identify the top 100 genes that were positively/negatively correlated with PC1, while accounting for the smooth spatial variation of the gene expression maps, we performed a spin test to derive significance of the correlation between each gene's expression pattern across and PC1 score across thalamic seeds. For genes showing a significant effect, we extracted the 100 with the strongest positive correlation and 100 with the strongest negative correlation. Because thalamic data is represented as a volume rather than a surface, a separate spin-test was applied using the BrainSMASH python toolbox

(version: 0.11.0; https://github.com/murraylab/brainsmash)[65]. In this method the values of the thalamic seeds are randomly permuted, whereupon variogram modelling is used to smooth and rescale the data as to impose the original spatial autocorrelation[65]. We repeated this procedure 1000 times for the PC1 scores and calculated the correlation between each of these permutations and expression of all 2228 genes across thalamic seeds. This distribution of correlations for each gene-PC1 pair was then compared to the corresponding empirical correlation to establish significance ($p_{spin}$ < 0.05). The top 100 positively and negatively correlated genes (as determined by correlation magnitude) which reached significance were then selected for further analysis. This same procedure was repeated separate for PC2 and PC3 in order to identify genes which showed significant correlations with those axes as well.

### Modelling gene expression trajectories

Using pre-processed PsychENCODE bulk tissue mRNA data (http://development.psychencode.org/)[68], we first identified genes that were both expressed in the thalamus and DE across time. Differential expression was determined through pairwise comparison over nine developmental windows to determine genes that where enriched either prenatally or postnatally (a minimum of three significant pairwise differences were needed for a gene to be considered enriched for at least one of these timepoints).

To model the trajectory of medial- and lateral-genes, we used generalised additive models as previously described[43]. Briefly, genes expression was modelled as a nonlinear function of age with sex and RNA integrity number acting as fixed effects, along with a random intercept to account for sample-specific variation. The nonlinear function was specified to use a natural cubic spline with four knots evenly spaces across the age span for smoothness. AIC and BIC were used to evaluate model performance. Age-corrected relative gene-expressed was then calculated using the residuals of the best-fit nonlinear mixed model. Gene trajectories were calculated using a combination of Python (version: 3.8.5), and the R (version: 3.6.0) libraries nlme (version: 3.1–161; https://rdrr.io/cran/nlme/) and mgcv (version: 1.8–41; https://rdrr.io/cran/nlme/).

### Reporting summary

Further information on research design is available in the Nature Portfolio Reporting Summary linked to this article.

## Data availability

All data for this project was obtained from open-source repositories. This included: diffusion and structural MRI data for 100 adults from the Human Connectome Project S1200 subjects data release (https://www.humanconnectome.org/study/hcp-young-adult/data-releases)[110–112]; the MNI152 template from FSL (version: 5.0.11; https://fsl.fmrib.ox.ac.uk/fsl/fslwiki/FSL)[133]; Melbourne subcortical atlas (https://github.com/yetianmed/subcortex)[95]; the Schaefer400 17 network atlas (https://github.com/ThomasYeoLab/CBIG/tree/master/stable_projects/brain_parcellation/Schaefer2018_LocalGlobal)[60]; voxelwise expression maps for 2228 genes from the Allen Human Brain Atlas (http://www.meduniwien.ac.at/neuroimaging/mRNA.html)[50,51]; list of 2413 genes showing elevated expression in the human brain (https://static-content.springer.com/esm/art%3A10.1038%2Fs41593-018-0195-0/MediaObjects/41593_2018_195_MOESM4_ESM.xlsx)[52]; list of the top 500 differentially expressed genes across the mouse thalamus and their associated PC1 scores (https://static-content.springer.com/esm/art%3A10.1038%2Fs41593-019-0483-3/MediaObjects/41593_2019_483_MOESM3_ESM.xlsx)[14]; Gene expression data for 19,419 genes across 213 regions in the Allen Mouse Brain Atlas (https://doi.org/10.5281/zenodo.4609603)[55,57,58]; Axonal tracing data for 213 regions in the Allen Mouse Brain Atlas (https://doi.org/10.5281/zenodo.4609603)[55–57]; NiFti volume and flat map for visualising Allen Mouse

Brain Atlas data in CCFv3 space (https://scalablebrainatlas.incf.org/mouse/ABA_v3#downloads; http://download.alleninstitute.org/publications/allen_mouse_brain_common_coordinate_framework/cortical_surface_views/ccf/annotation/)[58,124]; 72 brain maps/annotations from the *neuromaps* toolbox (https://netneurolab.github.io/neuromaps/)[61]; the fsaverage 164k vertex surface template from Free-Surfer (version: 5.3.0; https://surfer.nmr.mgh.harvard.edu/)[117]; nine measures of cortical organisation in the mouse (https://doi.org/10.6084/m9.figshare.7775684.v1; https://github.com/benfulcher/mouseGradients)[28]; a measure of mouse brain hierarchy (based on cortico-cortical, thalamo-cortical, and cortico-thalamic connections; https://github.com/AllenInstitute/MouseBrainHierarchy/)[59]; drop-seq analysis of 89,027 cells in the adult mouse thalamus from the Drop-Viz database (http://dropviz.org)[33]; list of genes with protein expression in the human thalamus from the Human Protein Atlas (version: 21.1; https://v21.proteinatlas.org/humanproteome/brain/thalamus)[128]; list of genes with identified mouse-human homologues from the Ensembl BioMart database (https://www.ensembl.org/index.html)[127]; and developmental gene expression from the PsychENCODE database (http://development.psychencode.org/)[68]. Data reported in this paper are provided in the Source Data file. Additionally, data generated and used in this study is openly available at https://doi.org/10.5281/zenodo.8285838. Source data are provided with this paper.

## Code availability

Code is available at https://github.com/StuartJO/ThalamicGradients.

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

## Acknowledgements

The authors would like to thank Martina Arenella and Mac Shine for helpful discussions and feedback on the manuscript. This research was supported by an NHMRC Investigator Grant (1194497 to G.B.), the Murdoch Children's Research Institute, the Royal Children's Hospital, Department of Paediatrics, The University of Melbourne and the Victorian Government's Operational Infrastructure Support Program. The project was generously supported by The Royal Children's Hospital Foundation devoted to raising funds for research at The Royal Children's Hospital. Data were provided [in part] by the Human Connectome Project, WU-Minn Consortium (Principal Investigators: David Van Essen and Kamil Ugurbil; 1U54MH091657) funded by the 16 NIH Institutes and Centres that support the NIH Blueprint for Neuroscience Research; and by the McDonnell Center for Systems Neuroscience at Washington University.

## Author contributions

S.O. and G.B. designed the research and wrote the manuscript. S.O. analysed the data, G.B. and S.O. performed the cell enrichment and gene trajectory modelling. G.B. supervised the project.

## Competing interests

The authors declare no competing interests.
