## [Peer Review File · Nature Communications]

A phylogenetically-conserved axis of thalamocortical connectivity in the human brainEditorial Note: This manuscript has been previously reviewed at another journal that is not operating a transparent peer review scheme. This document only contains reviewer comments and rebuttal letters for versions considered at *Nature Communications*.

Reviewer #1 (Remarks to the Author):

I appreciate the authors' response to my comments, especially their effort in addressing tractography results from the medial thalamus. I however think the concern is only partially addressed. I appreciate their normalization procedure in reducing medial bias, however, I think my main concern, the "accuracy" of thalamocortical tracking, is not sufficiently addressed. My main concern is that the tracking results are inaccurate (a mixture of false positives and false negatives), rendering the subsequent PCA less trustworthy. This is a somewhat well-known issue among those that study diffusion imaging. Thalamocortical projections are thin, and once they enter the central white matter area they are mixed with large cortico-cortical tracks, giving rise to false positives. The loss of signal in deep thalamic regions makes it difficult to track known projections that are well-studied in non-human primates (false negatives). I do not think the HCP data can address this inherent limitation. I think this issue must be addressed with better validation or clearly acknowledged (more so than the current draft) in the text.

Reviewer #2 (Remarks to the Author):

I would like to thank the authors for addressing my comments, and would like to congratulate them for this work.

Reviewer #3 (Remarks to the Author):

The authors did a great job of addressing each comment that was raised.

Reviewer #4 (Remarks to the Author):

I think the revised manuscript is greatly improved. The authors are to be commended on an extremely thoughtful revision. Very nice work.

We thank the Reviewers for their thoughtful comments. Our point-by-point response is provided below. Reviewer comments are in italics, our response is in normal font, and red text indicates text revisions.

Reviewer 1

I appreciate the authors' response to my comments, especially their effort in addressing tractography results from the medial thalamus. I however think the concern is only partially addressed. I appreciate their normalization procedure in reducing medial bias, however, I think my main concern, the "accuracy" of thalamocortical tracking, is not sufficiently addressed. My main concern is that the tracking results are inaccurate (a mixture of false positives and false negatives), rendering the subsequent PCA less trustworthy. This is a somewhat well-known issue among those that study diffusion imaging. Thalamocortical projections are thin, and once they enter the central white matter area they are mixed with large cortico-cortical tracks, giving rise to false positives. The loss of signal in deep thalamic regions makes it difficult to track known projections that are well-studied in non-human primates (false negatives). I do not think the HCP data can address this inherent limitation. I think this issue must be addressed with better validation or clearly acknowledged (more so than the current draft) in the text.

We appreciate the reviewers concerns and as a result have added the following text to the discussion:

We recognise that our estimates of connectivity along thalamic gradients may be affected by biases inherent to diffusion tractography including difficulty tracing from deep thalamic regions and the potential for false positive or negative connections. We note that tractography-derived thalamocortical connectivity broadly aligns with tract-tracing findings in primates^{34,84,85}, and we have attempted to mitigate this risk through validation of our human tractography results to those obtained using gold-standard tract-tracing in the mouse. Thus, while some finer details may be missed with tractography, we would expect the major patterns of connectivity we have described to be robust to specific tracking limitations. Nevertheless, addressing these limitations through new tractography techniques, improved MRI acquisitions or comparison to thalamocortical gradients in non-human primates using tract-tracing data represents an exciting direction for future research.

We wish to thank this and the other reviewers for their feedback on our manuscript.